# Safinamide in Clinical Practice: A Spanish Multicenter Cohort Study

**DOI:** 10.3390/brainsci9100272

**Published:** 2019-10-11

**Authors:** Gloria Martí-Andrés, Rayco Jiménez-Bolaños, José Matias Arbelo-González, Javier Pagonabarraga, Carmen Duran-Herrera, Rafael Valenti-Azcarate, Mª Rosario Luquin

**Affiliations:** 1Movement Disorders Unit, Clínica Universidad de Navarra (CUN), 31008 Pamplona, Spain; gmartia@unav.es (G.M.-A.); rvalenti@unav.es (R.V.-A.); 2IdiSNA (Navarra Institute for Health Research), 31008 Pamplona, Spain; 3Movement Disorders Unit, Hospital Universitario Insular de Gran Canaria, 35016 Las Palmas de Gran Canaria (HUIGC), Spain; rayco.jb.92@gmail.com (R.J.-B.); jmarbelo@gmail.com (J.M.A.-G.); 4Movement Disorders Unit, Hospital de la Santa Creu i Sant Pau (HSCSP), 08041 Barcelona, Spain; JPagonabarraga@santpau.cat; 5Movement Disorders Unit, Hospital Universitario de Badajoz, 06080 Badajoz, Spain; mariacarmenduranherrera@gmail.com

**Keywords:** Parkinson’s Disease, Safinamide, Adverse drug event, Dyskinesia, Drug-Induced, Motor complications

## Abstract

**Background:** Safinamide is an approved drug for the treatment of motor fluctuations of Parkinson’s Disease (PD) patients with a potential benefit on non-motor symptoms (NMS). **Methods:** A retrospective multicenter cohort study was conducted, in which the clinical effect of safinamide on both motor and NMS was assessed by the Clinical Global Impression of Change scale. Furthermore, we assessed the appearance of adverse events (AEs) and its effect on dyskinesia, that were also recorded in non-fluctuating PD patients and in those previously treated with rasagiline. **Results:** We included 213 PD patients who received safinamide in addition to their regular levodopa therapy. Thirty-five withdrew prematurely from safinamide, mainly because of AEs. Out of 178, clinical improvement on motor and NMS was found in 76.4% and 26.2%, respectively. A total of 44 reported AEs of mild intensity. We did not find a difference concerning the clinical benefit or AEs when comparing either patients who had or had not been taking Monoamine Oxidase B Inhibitor (MAOB-I) previously or between patients with and without motor complications. **Conclusions:** Safinamide is an effective and safe add-on to levodopa drug for PD patients. Moreover, safinamide could elicit an additional clinical improvement in PD patients previously treated with other MAOB-I and in non- fluctuating patients with suboptimal motor control.

## 1. Introduction

Parkinson’s disease (PD) is the second-most frequent neurodegenerative disorder after Alzheimer’s disease. Currently, it affects approximately 2–3% of the population above age 65. Since ageing is the most important risk factor for developing PD, its incidence and prevalence are expected to increase over the next decades [1]. A reduction of dopamine levels in the basal ganglia due to the progressive loss of Substantia Nigra pars compacta (SNpc) neurons, and the presence of deposits of alpha-synuclein protein called Lewy bodies (LB), have been classically described as the PD neuropathological hallmarks. However, PD is a complex and heterogeneous neurodegenerative disorder that involves multiple brain areas and neurotransmitter systems beyond the nigrostriatal dopaminergic pathway. Classically, PD is characterized by the well-known motor symptoms of the disease (bradykinesia, rigidity and resting tremor), but a wide range of Non-Motor Symptoms (NMS) can appear at any stage of the disease or even precede the motor manifestation [2,3]. Moreover, in the advanced stages of PD the combination of motor and NMS usually provokes a marked functional disability to patients, which urges the need for seeking an individualized treatment strategy.

Among the drugs approved for symptomatic treatment of PD, levodopa remains the gold standard. Nevertheless, long-term levodopa administration, disease progression and pulsatile drug delivery are considered important risk factors for the development of motor and non-motor complications [4]. Although several strategies have been used to treat or even delay the appearance of these motor complications [3,5,6,7,8], many recent advances have failed to manage these long-term levodopa side-effects.

Safinamide (Xadago^®^) is an orally administered α-aminoamide derivative recently approved for the treatment of mid- to late-stage fluctuating PD as an add-on therapy to a stable dose of levodopa alone or in combination with other PD treatments. Safinamide has a dual action mechanism. It is a potent, selective and reversible inhibitor of monoamino oxidase B (MAOB) and also modulates glutamate release. However, the role of this non-dopaminergic property in the global antiparkinsonian benefit elicited by the drug remains unknown. Although some studies indicate that the antiglutamatergic action appears with high doses, the recommended dosage of safinamide is 50–100 mg/day. Randomized controlled trials in mid-to late-stage fluctuating PD have showed not only a significant increment in daily ON time without troublesome dyskinesia, but also an improvement in clinical status (Clinical Global Impression-Severity of Illness (CGI-S),CGI-Change (CGI-C)), quality of life (PDQ-39) and Unified Parkinson’s Disease Rating Scale (UPDRS) part II-III-IV total scores [9,10,11]. Furthermore, post-hoc analysis of the pivotal studies and some open-label studies have found an improvement in NMS, such as PD-related chronic pain, mood and sleep disturbances [12,13,14,15]. All clinical trials have also showed that safinamide is well tolerated and produces a low incidence of Adverse Events (AEs), most being of mild-to moderate severity. In studies of clinical practice, patients aged 75 years or older and those with more severe stages of the disease were less likely to tolerate safinamide [16,17].

The goal of this study was to assess the effect of safinamide as an add-on to levodopa treatment by the CGI-C scale. Secondary outcomes were safety and tolerability of safinamide, and its effect on severity of dyskinesia.

## 2. Materials and Methods

### 2.1. Study Design and Population

We conducted an observational, retrospective, multicenter cohort study to assess the effectiveness (measured by CGI-C scale), safety and tolerability of safinamide in patients with diagnosis of PD. We included patients who fulfill UK Parkinson’s Disease Society Brain Bank Diagnostic Criteria [18] with no age restriction and who started safinamide therapy according to daily clinical practice. We excluded patients with insufficient medical data from clinical follow-up reports. All patients received safinamide as an add-on therapy to a stable dose of levodopa and should have been evaluated at least twice (baseline visit and first follow-up visit). The visit when safinamide was indicated was considered the baseline visit. The follow-up visit was defined as the first visit after the baseline visit, considering that both should be separated at least by 2 months.

All patients included were attended at the Movement Disorders Outpatient Clinics of 4 different centers. The recruitment period was between February 2016 and November 2017 (both included). We established a protocol to define the variables of interest to homogenize the data collection. Demographic and clinical data were systematically collected at the baseline visit, including falls, psychiatric disorders, cognitive impairment, presence of dyskinesia and motor-fluctuations, modified Hoehn and Yahr score, mean daily dose of levodopa (LD), treatment with dopamine agonists (DA), catechol-O-methyl transferase inhibitor (COMT-I), monoamine oxidase B inhibitor (MAOB-I), amantadine and their levodopa-equivalent dose (LEDD), among other variables.

The primary outcome was the clinical effect of safinamide on the motor and NMS separately, evaluated by the changes in CGI-C scale between the baseline and the follow-up visits. At the end of the study, CGI-C score was calculated as the sum of all available information, including a knowledge of the patient’s history, psychosocial circumstances, motor and non-motor PD-related symptoms, behavior and the impact of the symptoms on the patient’s ability to function. For this purpose, a movement disorders specialist made a judgment of change based on both information collected during clinical interview from patients and caregivers and neurological examination findings. The clinician asked systematically about motor symptoms (improvement on bradykinesia, muscular rigidity, resting tremor and gait) and NMS (cognitive function and attention level, sleep, neuropsychiatric and sensory disturbances) and compared the patient’s overall motor and non-motor clinical condition to the 1-week period just prior to the initiation of safinamide use. To simplify the statistical analysis and to lead to easy interpretation and presentation of the results, we clustered the CGI-score in 3 groups: improvement (substantial improvement (1), moderate (2) and minimum improvement (3)), no change (4) and worsening (minimum worsening (5), moderate (6) and substantial worsening (7)).

The secondary outcomes included the effect of safinamide on the severity of dyskinesia at the follow-up visit. Four movement disorder specialists were asked to classify them into 4 groups (improvement, no change, mild worsening (non-troublesome dyskinesia) and moderate-severe worsening (troublesome dyskinesia)). On the other hand, we evaluated the safety and tolerability of the drug by assessing the AEs presumably related to safinamide at the follow-up visit.

The medical ethical committee of the Universidad de Navarra approved the study, and all patients gave written informed consent.

### 2.2. Statistical Analysis

Quantitative variables were described by mean (standard deviation) if normally distributed, or by median (Interquartile Range, IQR) if not normally distributed. Distribution of the values was assessed by Shapiro–Wilk test. When normally distributed, values were compared by unpaired *t*-test; when not normally distributed, they were compared by Mann–Whitney U test. To compare dichotomic variables the chi-squared test was applied (or Fisher’s exact test, when appropriate). A logistic regression was used to assess the relationship between variables.

The statistical software Stata^®^ (Version 14.2) was used for all statistical analysis. *P* values < 0.05 were considered statistically significant.

## 3. Results

### 3.1. Subjects

We recruited 213 patients with PD who started safinamide according to daily clinical practice. There were 96 women (45.1%) with median age 68.6 years (IQR 60.9–74.5) and median disease duration 7.7 years (IQR 4.8–11.6). Baseline patients’ characteristics are summarized in Table 1.

Thirty-five patients (16.4 %) withdrew from safinamide treatment within the first month due to AEs in 24 patients, motor function deterioration in five, and lack of benefit on motor and non-motor function in six patients (further details are described in Section 3.3). This group of patients showed greater scores in the modified Hoehn & Yahr (mH&Y)scale, were receiving greater LEDD and showed higher cognitive impairment than those who continued the treatment (Table 2). Furthermore, higher mH&Y scores were independently associated with greater risk of safinamide withdrawal (OR 2.06; CI 95%:1.11–3.82; *p* = 0.021) after an adjustment for age, sex, disease duration, history of falls, cognitive impairment and LEDD.

One hundred and seventy-eight patients continued safinamide treatment for at least 2 months (84 women (Table 2)). In 66 patients (37.1%), the starting safinamide dose was 50 mg/day during the first month and then increased to 100 mg/day. In 82 patients (46.1%), the starting dose was 50 mg/day without further increment and in 30 patients (16.8%) safinamide was started directly at 100 mg/day. Patients on a safinamide final dose of 100 mg showed a more advanced disease stage than on a 50 mg final dose (higher disease duration, LEDD, cognitive impairment, history of dyskinesia and falls) (Table 3). The follow-up visit was performed at a median of 6 months (IQR 4.6–8.3) after baseline visit. Specifically, for patients who started on safinamide 50 mg and later increased to 100 mg the follow-up visit was performed at a median of 6.4 months (IQR 5.3–9) and for patients who started directly on 50 mg and 100 mg of safinamide was performed at a median of 6.1 (IQR 5.5–11.1) and 5.4 months (IQR 4–6) respectively. Patients started on safinamide remained on the same stable dosage of levodopa at least until the follow-up visit. A subgroup of 97 patients (54.5%) had been treated previously with rasagiline. In this group the decision of switching from rasagiline to safinamide was based on the presence of motor fluctuations (MF) and/or a suboptimal motor state. In these patients, we chose an overnight switch, and we did not observe a higher rate of AEs compared to the other patients (Table 2).

### 3.2. Efficacy

Concerning the effect of safinamide on motor and NMS as a whole (selecting the better score of both), 136 patients (76.4%) reported some improvement (CGI score 1–3), 31 patients (17.4%) reported no change (CGI score 4) and 11 described some worsening (5.5%) (CGI score 5–7). Focusing on motor symptoms, 136 patients (76.4%) referred some improvement (50 and 86 patients on safinamide 50 mg and 100 mg, respectively), 30 patients (16.8%) reported no change (13 and 17 patients on safinamide 50 mg and 100 mg, respectively) and 12 (6.7 %) denoted some worsening (all on safinamide 100 mg). Considering those patients who complained of NMS at baseline visit (145 out of 178 patients), 38 patients (26.2%) reported some improvement in NMS (13 and 25 patients on safinamide 50 mg and 100 mg, respectively), 102 patients (70.3%) noted no changes (44 and 58 patients on safinamide 50 mg and 100 mg, respectively) and five reported worsening of NMS (3.5%) (all on safinamide 100 mg). Patients on safinamide final dose of 100 mg showed higher probability of motor symptoms improvement but also of worsening than final dose of 50 mg (Fisher exact test, *p* = 0.019). Concerning NMS there were not statistically significant differences in CGI-C scores between final doses (Fisher exact test, *p* = 0.139). However, we did not find a significant relationship between global CGI score and the final dose of safinamide (OR 1.5, *p* = 0.247) after an adjustment for age, sex, disease duration, history of falls, cognitive impairment and LEDD.

After the follow-up visit, 139 patients (78.1%) did not require any further modification in their regular dopaminergic therapy and the dose of LEDD was maintained. In 17 (9.6%) patients, safinamide addition required a mild LEDD decrease, in 21 patients (11.8%) LEDD was increased despite maintaining safinamide treatment, and finally in one patient (0.5%), the dose of safinamide was reduced (final dose 50 mg) because of the worsening of dyskinesia after increasing the dose to 100 mg. There were different causes for the reduction of LEDD (155 mg (IQR 141–160 mg)). In 12 patients, adjunct safinamide therapy allowed us to reduce LEDD without any motor impairment. In four patients LEDD was reduced because of an increase in the intensity of dyskinesias and finally one patient developed a confusional state during the first month after starting safinamide that was successfully managed by reducing the levodopa dose. In the group requiring an increment of the LEDD for a better motor control, eight patients had not experienced any effect on motor symptoms. Although 13 patients reported a motor improvement, it was not good enough under the clinician’s discernment (Table 4).

### 3.3. Tolerability and Safety

The main reason for the premature abandonment of safinamide in 24 out of 35 patients was the appearance of AEs. Overall, 13 patients reported gastrointestinal disturbances, 8 a confusional state, 7 gait instability (not related to dyskinesia) and 2 worsening of dyskinesia. A total of 18 patients reported only one AE, 4 two types of AEs and only one patient suffered 3 types of AEs.

Considering the remaining patients (*n* = 178), 44 patients (24.7%) reported at least one AE, 22 (12.4%) reported worsening of dyskinesia, 18 (10.1%) gastrointestinal disturbances, 9 gait instability (5.1%), 3 confusional state (1.7%) and 1 intermittent blurred vision (0.5%) not related to a retinal disorder. Overall, 36 patients (20.2%) reported one AE, 7 patients (3.9%) two AEs and one patient (0.5%) three AEs.

Regarding the effect of safinamide on dyskinesia (*n* = 213), most patients showed no change in their intensity (182, 85.4%); in 7 patients (3.3%) safinamide reduced the intensity of dyskinesia, 14 patients (6.6%) showed a mild worsening in the severity of dyskinesia (2 “de novo” dyskinesia, 12 worsening of previous reported dyskinesia). Finally, 10 patients (4.7%) showed a moderate-severe worsening of dyskinesias (2 “de novo” dyskinesia, 8 worsening of previous reported dyskinesia).

### 3.4. Previous Treatment with MAOB-I (“MAOB-I” vs. “no MAOB-I”)

More than 50% of the patients were taking rasagiline before switching to safinamide (MAOB-I group *n* = 97 (54.5%) vs. No MAOB-I group *n* = 81 (45.5%), Table 5). There were no clinical differences between both groups of patients, although the MAOB-I group was slightly younger and showed lower scores in the mH&Y scale. Interestingly, the MAOB-I group showed an improvement in motor (80.4%) and NMS (32.5%) assessed by CGI-C scale after switching to safinamide.

### 3.5. Patients with or without Motor Complications (No MC vs. MC)

27 patients (15.2%) did not refer either MF or dyskinesia when they started safinamide. As expected, this PD group initiated safinamide treatment at earlier stages than the group with MC (*n* = 151) (Table 6). Thus, the disease duration, the mH&Y scores and the LEDD were also significantly lower in the non-fluctuating group. Interestingly, this group of patients displayed an improvement in motor (76.9%) and NMS (36.8%) assessed by CGI-C scale. Additionally, only 4 out of 27 (14.8%) described AEs after starting safinamide.

## 4. Discussion

This is the first observational multicenter Spanish study describing the effectiveness, tolerability and safety of safinamide in real clinical practice. Our results are consistent with previous studies, and confirm that safinamide can be considered an effective and safe add-on therapy for symptomatic PD treatment [9,10,11,16,17,19,20]. In general, safinamide was well tolerated with a dropout rate (DR) of 16.4%, which was even lower when considering only the AE-related DR (11.2%). Previous studies have reported very variable DR after starting safinamide, and this inconsistency is probably due to methodological differences in the study design (length of observation period, total or EA-related DR, etc.) [9,16,19]. For instance, in our cohort, the patients who withdrew from safinamide showed a more advanced stage of the disease (higher mH&Y scores, higher LEDD, cognitive impairment and a trend to be older) than those that continued safinamide treatment. All these data indicate that the severity of PD is an important factor in determining the tolerability to the drug and indicates that, as reported by Mancini et al. [16], the clinical profile of PD patients should be taken into account when prescribing safinamide to fluctuating PD patients.

The CGI-C scale was chosen to assess the efficacy of safinamide in real clinical practice because it is an easy and quickly applied research tool to quantify and track patient response to a drug. Several research drug efficacy scales are available for specific motor and non-motor features related to Parkinson’s Disease (UPDRS, PDQ-39, Hamilton Rating Scale for Depression, Hamilton Rating Scale for Anxiety, Montreal Cognitive Assessment test, etc.) which doubtless would provide a deeper and delimited assessment of patient response to the treatment. However, the CGI-C scale provides a whole impression of a patient’s overall clinical condition and is more feasible to apply in real clinical practice with multiple demands and limited time to deliver excellent care to patients and their caregivers. In our study, we found that more than 75% of patients showed a benefit on motor and/or NMS when safinamide was added to the conventional levodopa regimen and in a few patients LEDD could be reduced. This benefit was independent of the safinamide final dose after adjustment for confounding factors. These results confirm and extend previous data from randomized controlled trials and also those obtained in several open-label studies in which the addition of safinamide to conventional levodopa was followed by a significant improvement in motor and NMS as well as in the quality of life [16,19,20]. More importantly, our results indicate that the clinical benefit reported in controlled trials can be replicated in daily clinical practice. However contrary to the randomized studies, the levodopa dose was not significantly reduced in our cohort of PD patients [9,10,11], probably because of the relatively lower dose of levodopa taken by our PD patients when they started safinamide as compared to other studies.

Safinamide is a well-tolerated drug, especially in earlier stages of disease (lower mH&Y scores) and less than 25% of patients who maintained safinamide therapy reported AEs of mild intensity. As described in other studies, the most frequently AE found in our study was dyskinesia, [10,16,21]. However, it is important to note that most PD patients who complained of dyskinesia have previously presented these abnormal movements, indicating that safinamide might not induce dyskinesia in dyskinesia-naïve PD patients, although it may aggravate pre-existing dysknesia On the other hand, we did not find a relevant antidyskinetic effect in our PD patients as has been suggested in a previous study [21]. However, this observation should be taken with caution because our follow-up period was shorter (6 months vs. 24 months) and whether the antidyskinetic property is a progressive long-term effect is not yet known.

Finally, we have addressed two specific conditions that are relevant in clinical practice. On the one hand, we aimed to assess whether the switch of the most used MAOB-I (rasagiline) to safinamide can elicit an additional benefit. Although clinical scores to validate this additional benefit are not available, patients noted a significant clinical benefit on motor and/or NMS after switching from rasagiline. It is important to note these patients underwent an overnight switch to safinamide with no significant AEs. These data suggest that patients with a suboptimal control of motor and NMS who are taking rasagiline in combination with levodopa could obtain an additional clinical benefit from switching to safinamide. These data are supported by those observed by Mancini et al., in which PD patients taking other MAOB-I reported a significant reduction in daily OFF time and LEDD [16] by changing to safinamide. These results could be due to the broader pharmacodynamic profile of safinamide in relation to dopamine and glutamate pathways. However, further and well-designed prospective studies are required to confirm these observational data.

On the other hand, some of our patients did not suffer clear MF nor dyskinesia (15.2%, *n* = 27) before starting safinamide, but they reported a suboptimal motor function despite being treated with levodopa in combination with other dopaminergic drugs. For this reason and based on the results of pivotal trials of safinamide in early PD (015 and 017 studies) showing an improvement in activities of daily living and motor function (UPDRS part II-III) [22] and a reduction of rate of intervention in patients with stable motor response but suboptimal control [23], we decided to add safinamide to their regular therapy even though they did not exhibit MC.

When comparing PD patients with MC and those without MC, we found that patients without MC showed an earlier stage of the disease (lower disease duration and mH&Y scores) and were taking lower LEDD than PD patients with MC. Interestingly, patients without MC showed a significant clinical benefit and good tolerability when safinamide was added, suggesting that safinamide could be a useful therapeutic approach to treat PD patients with no MF but who require better motor control. However, it should be taken into account that MF can be underestimated by routine neurological clinical evaluation, in spite of being performed by a movement disorder specialist [24], and it is possible that some PD patients from this group were in fact mild-fluctuating patients. This fact supports the need to carry out clinical trials with safinamide in early and non-fluctuating patients.

Our observational retrospective cohort study has some limitations. We established a protocol to try to standardize data acquisition and excluded all patients without complete medical reports, but this does not preclude a selection bias. In addition, changes in safinamide dose were not predetermined; on the contrary, treatment decisions were based on severity of the disease and patient’s clinical situation. However, it is important to point out that it is a large, multicenter clinical practice study with a presumably high external validity.

## 5. Conclusions

In summary, our data support the finding that safinamide is an effective and safe add-on to levodopa therapy in PD and confirm an optimal patients’ adherence in real world scenarios. Also, a rapid switch from rasagiline to safinamide seems to be possible and safe. Furthermore, patients who are taking MAOB-I associated with levodopa and who are not clinically well controlled could benefit from switching to safinamide. Additionally, our data suggest that non-fluctuating patients with suboptimal ON state could benefit from adding safinamide.

## Figures and Tables

**Table 1 brainsci-09-00272-t001:** Demographic and clinical data at baseline visit (*n* = 213).

Age (years) ^a^	68.6 years (60.9–74.5)
Female (%) ^b^	96 (45.1%)
mH&Y score ^a^	2.5 (2–3)
1	5 (2.4%)
1.5	4 (1.9%)
2	84 (39.4%)
2.5	33 (15.5%)
3	69 (32.4%)
4	18 (8.5%)
5	0
Disease duration, years ^a^	7.7 (4.8–11.6)
History of falls ^b^	21 (12.6%)
History of dyskinesia ^b^	90 (45.3%)
MF ^b^	169 (79.3%)
Dyskinesia and/or MF ^b^	183 (85.9%)
CI associated to PD ^b^	18 (10.8%)
-MCI	16 (9.6%)
-PDD	2 (1.2%)
LEDD (mg)	700 (500–900)
-LD-LEDD ^a^ (%) ^c^	450 (300–600) (100%)
-DA-LEDD ^a^ (%) ^c^	210 (160–240) (67.1%)
-COMTI-LEDD ^a^ (%) ^c^	165 (132–206) (25.7%)
-Amantadine-LEDD ^a^ (%) ^c^	200 (100–200) (17.4%)
Previous treatment rasagiline ^b^	115 (54.0%)
Non-motor symptoms ^b^	173 (81.2%)

^a^ Median (interquartile rank); ^b^ Number of patients (relative frequency); ^c^ Relative frequency taking the treatment. Abbreviation: mH&Y = modified Hoehn and Yahr; MF = Motor fluctuations; CI = Cognitive Impairment; MCI = Mild Cognitive Impairment; PDD = Parkinson Disease Dementia; LEDD = Levodopa Equivalent Daily Dose; LD = levodopa; DA = dopamine agonists; COMTI = catechol-O-methyl transferase inhibitor.

**Table 2 brainsci-09-00272-t002:** Comparison of baseline characteristics between patients who withdrew from safinamide and those who maintained it.

	Maintain Safinamide (*n* = 178)	Safinamide Withdrawal (*n* = 35)	*p*-Value
Age (years) ^a^	67.4 (60.8–73.6)	74.2 (60.2–78)	0.0580 ^c^
Female (%)	84 (47.2%)	12 (34.3%)	0.161 ^d^
mH&Y score ^a^	2.5 (2–3)	3 (2–3)	0.0024 ^c^
1	5 (2.8%)	0	
1.5	4 (2.3%)	0	
2	75 (42.1%)	9 (25.7%)	
2.5	29 (16.3%)	4 (11.4%)	
3	53 (29.8%)	16 (45.7%)	
4	12 (6.7%)	6 (17.1%)	
5	0	0	
Disease duration, years ^a^	7.3 (4.3–11.0)	8.0 (5.8–13)	0.1812 ^c^
History of falls ^b^	21 (11.8%)	6 (17.1%)	0.5080 ^d^
History of dyskinesia ^b^	70 (39.3%)	20 (57.1%)	0.051 ^d^
MF ^b^	139 (78.1%)	30 (85.7%)	0.308 ^d^
CI associated to PD ^b^	13 (7.3%)	7 (20.0%)	0.0280 ^d^
LEDD (mg)	641 (500–864.5)	805 (670–1000)	0.0045 ^c^
Previous treatment rasagiline ^b^	97 (54.5%)	18 (51.4%)	0.739 ^d^
Non-motor symptoms ^b^	145 (83.1%)	32 (91.4%)	0.150 ^d^

^a^ Median (interquartile rank); ^b^ Number of patients (relative frequency); ^c^ Mann–Whitney test; ^d^ Chi-squared test. Abbreviation: mH&Y = modified Hoehn and Yahr; MF = Motor fluctuations; CI = Cognitive Impairment; LEDD = Levodopa Equivalent Daily Dose.

**Table 3 brainsci-09-00272-t003:** Comparison of baseline characteristics between patients with final doses of 50 mg/day vs. 100 mg/day.

	50 mg/day (*n* = 82)	100 mg/day (*n* = 96)	*p*-Value
Age (years) ^a^	67.2 (61.1–72.9)	68.0 (62.3–75.2)	0.1368 ^c^
Female (%)	41 (50%)	43 (44.8%)	0.4780 ^d^
mH&Y score ^a^	2 (2–3)	2.5 (2–3)	0.0566 ^c^
1	3 (3.7%)	2 (2.1%)	
1.5	4 (4.9 %)	0	
2	39 (47.5%)	36 (37.5%)	
2.5	13 (15.9%)	16 (16.7%)	
3	21 (25.6%)	32 (33.3%)	
4	2 (2.4%)	10 (10.4%)	
5	0	0	
Disease duration, years ^a^	5.9 (3.0–8.8)	9.2 (6–13)	0.0000 ^c^
History of falls ^b^	4 (4.8%)	12 (12.5%)	0.0200 ^d^
History of dyskinesia ^b^	22 (26.8%)	48 (50%)	0.0020 ^d^
MF ^b^	66 (80.5%)	72 (75.8%)	0.636 ^d^
CI associated to PD ^b^	2 (2.4%)	11 (11.5%)	0.0210 ^d^
LEDD (mg)	525 (405–742)	800 (560–950)	0.0000 ^c^
Previous treatment rasagiline ^b^	49 (59.8%)	46 (47.9%)	0.109 ^d^
Non-motor symptoms ^b^	64 (78%)	76 (79.2%)	0.871 ^d^

^a^ Median (interquartile rank); ^b^ Number of patients (relative frequency); ^c^ Mann–Whitney test; ^d^ Chi-squares test. Abbreviation: mH&Y = modified Hoehn and Yahr; MF = Motor fluctuations; CI = Cognitive Impairment; LEDD = Levodopa Equivalent Daily Dose.

**Table 4 brainsci-09-00272-t004:** Management decisions after follow-up visit.

Management Decision	Management Justification
LEDD + safinamide maintained (*n* = 139; 78.1%)	-Good clinical status
LEDD decreased + safinamide maintained (*n* = 17; 9.6%)	-Reduction of LEDD without motor impairment (*n* = 12) -Increase of intensity of dyskinesia (*n* = 4) -Confusional state (*n* = 1)
LEDD increased + safinamide maintained (*n* = 21; 11.8%)	-No effect on motor symptoms (*n* = 8) -Unsatisfactory motor improvement (*n* = 13)
LEDD maintained + safinamide decreased (*n* = 1; 0.5%)	-Increase of intensity of dyskinesia with 100 mg/day (*n* = 1)

**Table 5 brainsci-09-00272-t005:** Comparison between patients with or without a previous MAOB-I.

	MAOB-I (*n* = 97)	No MAOB-I (*n* = 81)	*p*-Value
Age (years) ^a^	65.2 (58.3–71.9)	70.0 (62.2–76.2)	0.0015 ^c^
Female (%) ^b^	46 (47.4%)	38 (46.9%)	0.9460 ^d^
mH&Y score ^a^	2 (2–3)	2.5 (2–3)	0.0034 ^c^
1	3 (3.1%)	2 (2.4%)	
1.5	2 (2.1 %)	2 (2.4%)	
2	53 (54.6%)	22 (27.2%)	
2.5	12 (12.4%)	17 (21.0%)	
3	21 (21.7%)	32 (39.5%)	
4	6 (6.2%)	6 (7.4%)	
5	0	0	
Disease duration, years ^a^	7.0 (4.2–10.4)	7.9 (4.7–12.0)	0.4341 ^c^
History of dyskinesia ^b^	37 (38.1%)	33 (40.7%)	0.7240 ^d^
MF ^b^	81 (83.5%)	58 (71.6%)	0.0560 ^d^
CI associated to PD ^b^	8 (8.2%)	5 (6.7%)	0.5960 ^d^
LEDD (mg)	600 (500–820)	700 (450–900)	0.3722 ^c^
Non-motor symptoms ^b^	77 (79.4%)	65 (80.3%)	0.886 ^d^
Adverse events/Dyskinesia ^b^	21 (21.6%)	22 (27.2%)	0.731 ^d^
Improvement motor symptoms (CGI-C ≤ 3) ^b^	78 (80.4%)	59 (72.8%)	0.2320 ^d^
Improvement non-motor symptoms (CGI-C ≤ 3) ^f^	26/80 (32.5%)	12/65 (18.5%)	0.4100 ^d^

^a^ Median (interquartile rank]; ^b^ Number of patients (relative frequency); ^c^ Mann–Whitney test; ^d^ Chi-squared test. ^f^ Relative frequency out of total of patients with non-motor symptoms. Abbreviation: mH&Y = modified Hoehn and Yahr; MF = Motor fluctuations; CI = Cognitive Impairment; LEDD = Levodopa Equivalent Daily Dose; CGI-C = Clinical Global Impression of Change.

**Table 6 brainsci-09-00272-t006:** Between patients with Motor Complications (MC) and without MC.

	MC (*n* = 151)	No MC (*n* = 27)	*p*-Value
Age (years) ^a^	68.6 (61.0–73.9)	65.2 (56.0–69.5)	0.1048 ^c^
Female (%) ^b^	75 (49.6%)	9 (33.3%)	0.117 ^d^
mH&Y score ^a^	2.5 (2–3)	2 (2–2)	0.0001 ^c^
1	2 (1.3%)	3 (11.1%)	
1.5	1 (0.6 %)	3 (11.1%)	
2	59 (39.1%)	16 (59.2%)	
2.5	27 (17.8%)	2 (7.4%)	
3	50 (33.1%)	3 (11.1%)	
4	12 (7.9%)	0	
5	0	0	
Disease duration, years ^a^	8 (5.4–12.0)	2.7 (1.8–6.0)	0.0001 ^c^
CI associated to PD ^b^	13 (8.6%)	0	0.1130 ^d^
LEDD (mg)	700 (510–900)	500 (400–555)	0.0001 ^c^
Non-motor symptoms ^b^	126 (83.4%)	19 (70.4%)	0.1070 ^d^
Adverse events/Dyskinesia ^b^	39 (25.8%)	4 (14.8%)	0.218 ^d^
Improvement motor symptoms (CGI-C ≤ 3) ^b^	126 (76.7%)	19 (76.9%)	0.977 ^d^
Improvement non-motor symptoms (CGI-C ≤ 3) ^b^	31/126 (24.6%)	7/19 (36.8%)	0.5040 ^d^

^a^ Median (interquartile rank); ^b^ Number of patients (relative frequency); ^c^ Mann–Whitney test; ^d^ Chi-squared test. Abbreviation: mH&Y = modified Hoehn and Yahr; MF = Motor fluctuations; CI = Cognitive Impairment; MCI = Mild Cognitive Impairment; PDD = Parkinson Disease Dementia; LEDD = Levodopa Equivalent Daily Dose. CGI-C = Clinical Global Impression of Change.

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
