# Peer review of "Safinamide in Clinical Practice: A Spanish Multicenter Cohort Study"

_brainsci, 2019, doi:10.3390/brainsci9100272_

Round 1

Reviewer 1 Report

In this study, Marti-Andres and his coauthors examine the value of safinamide in clinical practice in the Spanish PD patient population. Designed as a retrospective unblinded multicenter (n = 4) cohort study, they evaluate clinical effects of safinamide on motor and non-motor symptoms (NMS) as well as adverse events (AE).

The patient population is includes a rather small number of 213 PD patients who received safinamide in addition to their regular levodopa therapy and were treated in specialized movement disorder centers in Spain between February 2016 and November 2017.

The primary outcome was the clinical effect of safinamide on the motor and NMS separately evaluated by changes in the CGI-C scale between the baseline and the follow-up visits, which was at least 2 months later. The secondary outcomes included effect of safinamide on severity of dyskinesia at the follow-up visit. The methodological analysis has some severe flaws which are described below as major points.

35 patients withdrew prematurely because of AEs. Of the remaining 178 patients 76.4 % reported an improvement of motor function and 26.2 % were better in NMS. There was no difference in patients who had an additional treatment with MAO-B inhibitors. The authors conclude that Safinamide is an effective and safe add-on to levodopa drug for PD and can elicit an additional clinical improvement in PD patients previously treated with other MAO-B inhibitors.

The manuscript is well written and clearly structured. The results add more real-world data on the use of safinamide in clinical practice, now for the Spanish population. Similar studies have been performed in other countries (e.g. WH Jost and colleagues 2018 in Germany with a follow-up of 6 months, MLE Bianchi and colleagues 2018 in Italy with a follow-up of at least 3 months), which basically replicate the findings of the pivotal studies 016 and 018 by Borgohain and colleagues from 2014, and the study by Schapira et al. in 2017 JAMA Neurology.

The authors themselves state that their results confirm previous data from randomized controlled trials. One novel aspect is of course the examination of a Spanish real-world population. The data on the switch from rasagiline to safinamide are important to know, as these do not indicate any major problems if the switch is done overnight. As limitations of the study, there are some major methodological points, which have to be addressed:

The methodology to determine the CGI-C (clinical global impression of change) is not clear. In the methods section it is stated that “Finally, the definitive (CGI-C) score was obtained considering both the subjective impression given by patient/caregiver andthe objective neurological examination findings.” How exactly was the “definitive” CGI-C score calculated ? What percentage was contributed by the patients impression and what percentage by the clinicians impression or was is just a “best fit” ? Patients were started on safinamide 50 mg and later increased to 100 mg orleft on 50 mg orwere directly started on 100 mg. These are rather heterogeneous groups which are not further analyzed or discussed concerning outcome measures. It has to be explicitly stated that patients started on safinamide remained on the same “stable” dosage of levodopa, meaning that the levodopa dosage was NOT changed while under safinamide from its start to follow-up. the patient observation time is rather short with only at least 2 months. What is the average observation time for the different patient groups ?

Author Response

Dear Editor,

Thank you very much for considering our manuscript for publication in Brain Sciences, section Clinical Neuroscience.

I very much appreciate the comments of the reviews which I am sure will serve for improving the manuscript.

Below you will find our responses to every issue that the reviewers have suggested.

Sincerely yours.

Reviewer 2 Report

Decent multicenter study may provide extra clinic data to support a relatively new drug " Safinamide" for improving the therapeutic efficacy of Parkinson's disease. The design and conclusion is adequate and reasonable. The paper addressed the several key clinic concerns about the usage of a new drug for PD like " Saftey" " Tolerability" " Efficacy" " previous MAOB-I use" etc.

Some minor typo or errors need recheck and edit. For example" at line 90, "LD-equivalent dose ( LEDD) " should be " Levodopa-equivalent daily dose (LEDD)". 

In the discussion part, it would be better or more convincing if the author can discuss more regarding "why use CGI-C as a measurement instead of other scales", what  are the difference among those different scales.

Author Response

(The authors gave the same response as above.)

Round 2

Reviewer 1 Report

The major points of the former review have all been adequately addressed. This has considerately improved the manuscript.